# QRS Narrowing Following CRT Implantation: Predictors, Dynamics, and Association with Improved Long-Term Outcome

**DOI:** 10.3390/jcm11051279

**Published:** 2022-02-26

**Authors:** Daniel Lapidot, Moshe Rav-Acha, Tali Bdolah-Abram, Rivka Farkash, Michael Glikson, Tal Hasin

**Affiliations:** 1Faculty of Medicine, Hebrew University, Jerusalem 9103102, Israel; danielap727@gmail.com (D.L.); ravacham@szmc.org.il (M.R.-A.); taliba@savion.hji.ac.il (T.B.-A.); mglikson@szmc.org.il (M.G.); 2Jesselson Integrated Heart Center, Shaare Zedek Medical Center (SZMC), Jerusalem 9103102, Israel; rivka_f@szmc.org.il

**Keywords:** cardiac resynchronization therapy, QRS duration, QRS narrowing, outcome

## Abstract

Background: Heart failure (HF) patients with wide QRS often benefit from cardiac resynchronization therapy (CRT), although QRS narrowing does not always occur. The current study investigates the incidence and predictors for QRS narrowing following CRT and its long-term impact on clinical outcomes. Methods: Among individuals undergoing clinically indicated CRT, pre-and post-implantation electrocardiographs were meticulously analyzed for QRS duration change. All-cause mortality and the composite of mortality and HF hospitalizations were retrieved. Results: For 104 patients, mean age 67 years, 25% females, QRS narrowed within days by 20.2 ± 24.7 ms. In 55/104 (53%) QRS narrowed by ≥20 ms (“acute narrowing”). Female gender and baseline QRS predicted acute narrowing. Acute narrowing persisted for 1–6 weeks in 18/20 (90%) and 3–12 months in 21/31 (68%) of patients. During the average follow-up of 41 months, 29/104 (28%) died and 50/104 (48%) met the composite outcome. In a multivariable analysis including comorbidities and cardiac history, prolonged baseline PR interval (HR 1.015, CI 1.008–1.021, *p* < 0.001) and acute narrowing < 20 ms (HR 3.243, CI 1.593–6.603, *p* = 0.001) were significant and independent predictors for the composite outcome. Conclusions: Post-CRT acute QRS narrowing ≥ 20 ms is independently associated with favorable long-term outcomes and might be considered as a novel measure for procedural success.

## 1. Introduction

Cardiac resynchronization therapy (CRT) is used to treat patients with heart failure (HF) and left ventricular systolic dysfunction (LVSD). These patients often have regions of delayed myocardial activation and contraction, leading to cardiac desynchrony. In a series of trials, CRT was found to be beneficial in reducing HF symptoms, as well as in improving exercise capacity, quality of life, and left ventricular function [1,2]. A significant decrease in all-cause mortality after CRT was demonstrated in symptomatic HF patients with LVSD and cardiac electrical desynchrony (CARE-HF) [3]. Implantable cardioverter-defibrillators (ICDs) are used to prevent arrhythmia-related sudden cardiac death and the addition of CRT to an ICD was shown to improve survival and HF hospitalizations [4].

According to current ESC HF guidelines [5], indications for CRT implantation include symptomatic HF patients with LBBB morphology, QRS duration (QRSd) > 130 ms and EF < 35%, in which CRT was found to be superior to optimal medical treatment or to ICD alone. The importance of QRS duration and morphology was established by a few studies, [6,7] based on the MADIT-CRT [8] and RAFT [4] trials. Baseline QRSd > 130 ms, especially longer than 150 ms, and LBBB morphology predicts CRT response, as demonstrated in a meta-analysis of 5 major clinical trials [9]. Sub-analyses from randomized clinical trials suggested that the beneficial effects of CRT may be greater in specific subgroups such as women and non-ischemic HF patients [10,11].

Nevertheless, about 30% fail to show symptomatic improvement following CRT implantation [12]. Several unsuccessful attempts were made to predict non-responders through various imaging techniques [13,14]. Despite excessive research on background factors affecting CRT response in HF patients, there is limited data regarding the impact of acute QRS narrowing on long-term outcomes following CRT implantation. QRS narrowing may reflect the correction of cardiac asynchrony, and thus can be utilized as an early indicator of reverse LV electrical remodeling [15,16]. Hence, measurement of QRS narrowing following implantation may assist in predicting response to treatment.

The aim of this study is to examine the incidence, possible predictors, and prognostic impact of QRS narrowing following CRT implant. We also tried to assess the dynamics of post CRT QRS duration over time.

## 2. Methods

A retrospective analytical study of HF patients implanted with a CRT device (with and without defibrillator) from 2006 to 2020 in Shaare Zedek Medical Center (SZMC) cardiology department.

### 2.1. Inclusion and Exclusion Criteria

Inclusion criteria were symptomatic HF (NYHA II-III); moderate-severe or severe LV dysfunction by pre-implantation Echocardiography; and pre-implantation QRS duration ≥ 130 ms. Exclusion criteria included: (a) lack of at least 1 documented ECG within 6 weeks before CRT implantation and at least 1 ECG’s post implant; (b) CRT bi-ventricular pacing < 90%; (c) upgrade to a CRT device from a prior PPM or ICD device with >1% of ventricular pacing (VP). The study was conducted according to the guidelines of the Declaration of Helsinki and approved by the Institutional Ethics Committee of Shaare Zedek Medical Center (approved 29.4.2019; code 0099-19-szmc).

### 2.2. Follow-Up and Outcomes

Data regarding patients’ demographics and medical history were acquired from the CRT implantation hospitalization records (considered as “index” hospitalization). HF etiology was categorized as ischemic (based on either a history of myocardial infarction or a coronary angiogram test with significant stenotic lesions) or non-ischemic. Echocardiographic parameters (LV end-diastolic diameter, LV end-systolic diameter, LV function and fractional shortening) were documented based on two tests which were performed prior (up to 1 year) and following (first after) the implantation. Improved LV function was defined as categorical change (in a scale of normal; mild; mild-moderate; moderate; moderate-severe; severe) in LV function. Pre-implantation electrocardiographic variables included heart rate, rhythm, PR interval, QTc interval, QRS morphology and QRS duration. Electrocardiographic data were evaluated prior (within 6 weeks pre-CRT implant) and post-CRT implantation.

Primary outcomes were defined as death from any cause, and the composite of death from any cause or HF hospitalization. HF hospitalization was defined as such if HF exacerbation was the main reason for admission (exclusively at SZMC). Follow-up was achieved by analysis of all available medical documentation including device clinic reports. Data regarding mortality events were acquired from the Israeli ministry of interior affairs with full accessibility.

### 2.3. QRS Morphology and Duration Measurement

Electrocardiographic parameters were measured by using a standard 12-leads electrocardiogram (ECG).

Left bundle branch block was diagnosed using conventional criteria: QRS duration > 120 ms, QS/rS morphology in V1 and broad notched or slurred R wave in leads I, AVL, V5, V6 with R peak time > 60 ms in leads V5, V6 in accordance with AHA/ACC recommended definition of LBBB [17]. Otherwise, the QRS morphology was classified as “other”. QRS morphology, heart rhythm and rate, PR and QTc values were documented based on the last ECG prior to implantation.

Assessment of QRS duration (QRSd) was performed using the global method from the earliest onset to the latest offset waveform in all 12 ECG leads (as recommended by AHA/ACC ECG standardization document) [17], using digital calipers. In order to improve the accuracy of baseline QRSd prior to implantation—an average of up to three separate ECGs preformed within 1 year before the implant was calculated (Pre-QRS average). Post-implantation electrocardiographic data were considered within the following intervals—upon discharge from “index” hospitalization (“Acute”), 1–6 weeks following implantation (“Early”) and 3–12 months following implantation (“Late”). The difference between pre-implantation QRSd to each of these values was defined as Acute/Early/Late QRS narrowing. Notably, post-implantation QRS was evaluated during continuous bi-ventricular CRT pacing. In addition to the absolute QRS narrowing (ms), relative QRS narrowing (%) was calculated as the percentage of measured narrowing from the average baseline QRSd prior to implantation. Most (~90%) ECGs were evaluated during sinus rhythm.

### 2.4. Statistical Analysis

Descriptive statistics were displayed for qualitative (frequencies, %) and quantitative variables (Mean ± SD). For the identification of baseline variables that correlate with Acute QRSd narrowing ≥ 20 ms, Chi-square and test-test were done for qualitative and quantitative variables, respectively. Variables correlated with acute QRSd narrowing were included in a forward stepwise multivariable logistic regression model (including age and gender).

For examining the association between outcome measures and acute QRSd narrowing, a cox analysis was performed. Kaplan-Meier curve was used to investigate the effect of acute QRSd narrowing with a cutoff of 20 ms on survival and the composite endpoint.

The influence of baseline variables on survival, together with QRSd narrowing, was evaluated by cox regression model with forwarding stepwise multivariable analysis.

All statistical tests were 2-tailed, and a *p-value* of 0.05 was considered statistically significant. The software used for data analysis was SPSS (version 25).

## 3. Results

A total of 260 CRT implantations were documented between 2006–2020 in SZMC, of which 97 were excluded due to upgrade from a prior PPM or ICD devices with significant (>1%) VP. Of these, 37 were excluded due to CRT implanted without standard guidelines given baseline QRS duration < 130 ms (*n* = 21), or presence of mild/moderate LVSD (*n* = 16); 21 cases were excluded due to missing ECG data, and 5 excluded due to CRT biventricular pacing <90%. Accordingly, 104 cases were included in our study (Figure 1). Among the participants, 78/104 (75%) were males, mean age was 67 ± 11.7 years, 59/104 (57%) had ischemic etiology and 61/104 (59%) had severe LV dysfunction. Only 4/104 (4%) had prior ICD devices, and CRTD (rather than CRTP) was implanted in 98/104 (94%) (Table 1).

An average of 2.3 pre-CRT ECGs per patient were evaluated for baseline QRS validation. The average QRSd prior to implantation was 151.6 ± 14.3 ms. LBBB was identified in 97/104 (93%), average heart rate was 73 ± 13 bpm, the average PR interval was 198 ± 50 ms, and the average QTc was 472 ± 50 ms (Table 1).

The average time between CRT implantation and discharge ECG was 2 days. Mean absolute acute QRS narrowing was 20.2 ± 24.7 ms (Median 20.3 ms) and the relative acute QRS narrowing was 12.9 ± 13.5%. A cutoff of 20 ms for the definition of “acute QRS narrowing” was set based on the distribution of acute QRS narrowing among the study population (Figure 2), revealing 55/104 and 49/104 patients with and without acute QRS narrowing > 20 ms, respectively. Among the 49/104 without QRSd narrowing of >20 ms, 25 patients had QRSd narrowing of <20 ms and 24 patients had QRSd prolongation. Early (1–6 weeks) and late (3–12 months) electrocardiographic parameters were available for 31/104 (29.8%) and 64/104 (61.5%) patients, respectively. Of the patients with acute QRS narrowing ≥ 20 ms, the QRS narrowing persisted 1–6 weeks and 3–12 months following CRT implantation in 18/20 (90%) and in 21/31 (68%) of patients with documented early and late ECG, respectively. Among the patients who had acute QRS narrowing **<** 20 ms (or no acute QRS narrowing), a QRS narrowing ≥ 20 ms appeared during 1–6 weeks and 3–12 months post implantation in 4/11 (36%) and 11/33 (33%) patients with documented early and late ECG, respectively.

Baseline variables associated with acute QRS narrowing ≥ 20 ms were: female gender, no prior coronary interventions (PCI and CABG), and increased baseline QRSd (Table 1). In a multivariate analysis female gender (HR 4.454, CI 1.521–13.046, *p* = 0.006) and prolonged baseline QRSd (HR 1.068, CI 1.031–1.106, *p* < 0.001 per 1 ms increase in baseline QRSd) remained significantly associated with acute QRS narrowing ≥ 20 ms (Table 1). A possible interaction between the female gender and ischemic etiology was interrogated. Prior MI was strongly associated with males (*p* < 0.001). An exploratory analysis excluding gender showed a statistically significant negative association with adjusted OR of 3.46 (CI 1.18–10.15) between CABG and acute QRS narrowing < 20 ms.

During an average follow-up of 41 months, 29/104 (28%) died and 50/104 (48%) met the composite outcome of death or HF hospitalization. In a univariate cox analysis, acute relative QRS narrowing (% of baseline QRSd reduction) as a continuous variable was significantly associated with all-cause mortality (HR 0.974, CI 0.958–0.991, *p* = 0.002, Pearson Correlation = −0.247), and with combined all-cause mortality or HF hospitalization (HR 0.979, CI 0.966–0.992, *p* = 0.002) for every 1% decrease in QRSd. Acute QRS narrowing < 20 ms was also significantly associated with all-cause mortality (*p* 0.037) as well as with combined mortality and HF hospitalization (*p* 0.05). Kaplan-Meier curves for all-cause mortality and for the combined endpoint of all-cause mortality/HF hospitalization, according to acute QRS narrowing < or ≥20 ms, are shown in Figure 3. Patients with acute QRS narrowing ≥ 20 ms were found to have a significantly better prognosis. In a multivariable cox utilizing the stepwise approach (including all significant variables, as well as age and gender), prior valve surgery and history of MI were associated with increased mortality with HR of 3.39 (CI 1.35–8.48, *p* = 0.009) and 2.81 (CI 1.22–6.50, *p* = 0.016) respectively. In a multivariate analysis for the combined outcome of all-cause mortality or HF hospitalization, both prolonged baseline pre-CRT intrinsic PR interval (HR 1.01, CI 1.01–1.02, *p* < 0.001 per 1 ms increase), and acute QRS narrowing < 20 ms (HR 3.243, CI 1.593–6.603, *p* = 0.001) were independently associated with increased occurrence of the composite outcome (Table 2).

## 4. Discussion

The current study examines the impact of acute QRS narrowing following CRT implantation on survival and HF hospitalization. Among 104 patients, 93% with LBBB morphology, 53% displayed acute QRS narrowing ≥ 20 ms in sequential ECG within 2 days following implantation. Most (~70%) of patients with acute QRS narrowing, continued to have narrowed QRS up to 12 months after CRT implant. Predictors for acute QRS narrowing ≥ 20 ms were female gender, wider baseline QRSd and non-ischemic etiology. During an average follow-up of 41 months, 28% died and 48% met the composite outcome of mortality or HF hospitalization. Acute QRS narrowing < 20 ms was independently associated with the composite outcome (HR 3.243, CI 1.593–6.603, *p* = 0.001). Notably, around half of our study CRT recipients had ischemic cardiomyopathy, which is higher than reported in other CRT cohorts [18,19], a fact that might have contributed to the relatively high mortality rate among our CRT cohort.

In the current study female gender, wider baseline QRSd and non-ischemic etiology were found to predict acute QRS narrowing post CRT implant. At least to our knowledge, predictors for QRS narrowing post CRT have not been evaluated previously and are one of the novel findings of our work. Importantly, these same factors were found to predict overall favorable response to CRT implantation in the large CRT pivot trials [5,10,11]. It is therefore possible that these parameters predict overall CRT benefit due to their association with QRS narrowing, reflecting restored cardiac synchrony. If so, acute QRS narrowing might be the critical mechanism, underlying CRT clinical response, and might serve as a novel clinical procedural endpoint.

In addition to QRS narrowing, we found shorter baseline pre-CRT intrinsic PR interval to be significantly associated with favorable post-CRT outcomes. Previous studies [20,21,22] have pointed to a possible correlation between shorter baseline PR interval and better prognosis following CRT implantation. Interestingly, in contrast with QRS narrowing characterizing CRT responders, post-CRT PR interval tends to prolong, regardless of CRT response [22]. Further research is thus needed to examine the prognostic effects of monitoring post-implantation changes in intrinsic PR interval.

Acute QRS narrowing ≥ 20 ms was independently associated with the composite outcome of HF hospitalization or mortality. This result supports previous findings regarding the prognostic value of acute QRS narrowing. Notably, several prior studies [16,18,23,24,25,26] have evaluated the association of post-CRT QRS narrowing with the outcome. However, most of these studies evaluated the short-term association of QRS narrowing with “soft” endpoints such as echocardiographic response, mitral regurgitation severity or peak oxygen consumption. Interestingly, QRS narrowing > 20 ms post-CRT implant was found to predict CRT responders in a previous study, although with a short 6 months follow-up period [26]. Only a few prior studies evaluated the association of QRS narrowing with overall mortality, revealing improved survival during a median follow-up of 2–4 years [19,27,28]. A recent study by Jastrzębski et al. [27] showed that immediate QRS narrowing after CRT implantation was associated with a favorable prognosis, albeit only in patients with LBBB morphology. Similar to our study, Jastrzębski et al. showed QRS narrowing post CRT procedure among 72% of CRT recipients and the average QRS narrowing was 16 ms. However, while they showed a linear relation between QRS narrowing to improved survival, our study emphasizes the importance of a 20 ms QRS narrowing cutoff value for improved prognosis. Although this does not necessarily contradict, we believe it serves a more practical measure since < 20 ms QRS width differences are difficult to assess and are easily missed even by experienced cardiologists [27], and could at times be attributed to “noise”. Lastly, a recent study evaluating 3-year composite outcome including all-cause death, left ventricular assist device implantation, cardiac transplantation and HF hospitalization, found post CRT QRS shortening > 20 ms to predict the improved composite outcome [28], supporting the 20 ms cutoff.

This study is the first to investigate the dynamic nature of QRSd post CRT implant. A dynamic delayed QRS prolongation was previously reported in HF patients with initial narrow QRS [29]. Thus, it is reasonable to assume that QRSd may have a dynamic nature, especially after CRT implant. It is reassuring that QRS narrowing is persistent in most of the cases evaluated. Delayed narrowing in some patients without acute QRS narrowing necessitates further research and an evaluation of its prognostic impact. Notably, in our study, there was a non-significant trend for reduced composite outcome events among patients without acute QRS narrowing who developed such narrowing later on as compared to those who did not narrow their QRS during the following months as well (data not shown). However, given the limited number of patients who had regular ECG done early or late post CRT implant, follow-up ECG results need to be validated by future studies.

Alongside standard clinical indications for CRT implant, several imaging techniques aimed to evaluate cardiac desynchrony have failed to predict CRT response, as demonstrated in the PROSPECT trial [13]. However, based on our findings, measuring “acute” QRS narrowing post CRT implant can help identify patients who are less likely to respond to CRT. These patients may require frequent monitoring and possibly benefit from the optimization of CRT pacing. Currently, routine AV and VV delay optimization following CRT implantation is not recommended due to its limited effect on clinical or echocardiographic outcomes [5]. However, its utility may become evident among the subgroup of CRT non-responders. Accordingly, we suggest that AV and VV “electrical optimization” to minimize QRSd, may be warranted in patients who fail to show acute QRS narrowing following CRT implant. Furthermore, although in our study we evaluated QRS width at discharge and not during the CRT implant procedure, we suggest >20 ms QRS narrowing may serve as a “gold standard” clinical endpoint for successful CRT procedures. Notably, we present a procedural “clinical endpoint” rather than a classic pre-procedural predictor to predict which patients would eventually benefit from the CRT procedure. According to our suggested procedural clinical endpoint, an optimal LV lead location during the CRT procedure would be a location that will enable biventricular pacing to achieve QRS narrowing > 20 ms compared to the initial pre-CRT QRS width. Anyhow, since QRS narrowing in this study was not measured within the CRT procedure, this should be confirmed by a future study examining the impact of QRS narrowing at the end of the CRT procedure on the outcome.

### Strengths and Limitations

The strengths of this study include the strict inclusion criteria and the meticulous verification of baseline QRS duration including measurements in several time intervals. Our main limitations are the small sample size, and missing data, primarily post-implantation echocardiography and long-term electrocardiograms, derived from the retrospective nature of the study. Moreover, due to the retrospective nature of the study, device programming was not homogenous. Nevertheless, our study presents the results of a real-world cohort of CRT patients with all its strengths and limitations.

## 5. Conclusions

Although QRS is narrowed in most of the patients after CRT implant, only half of the patients shorten their QRSd by ≥20 ms within a few days post CRT implant (“acutely”). QRSd following CRT implant is dynamic, and although most patients with acute QRSd narrowing will continue to have narrowed QRSd during the following months this is not the case in all. Acute QRS narrowing is predicted by the same traditional parameters which were shown to predict overall CRT success (female gender, wide baseline QRS, and non-ischemic etiology) and is independently associated with improved outcomes. We suggest QRS narrowing > 20 ms as a possible acute procedural endpoint to predict improved long-term outcomes, and interventions to improve outcomes in patients without such acute QRS narrowing should be tested in future studies.

## Figures and Tables

**Figure 1 jcm-11-01279-f001:**
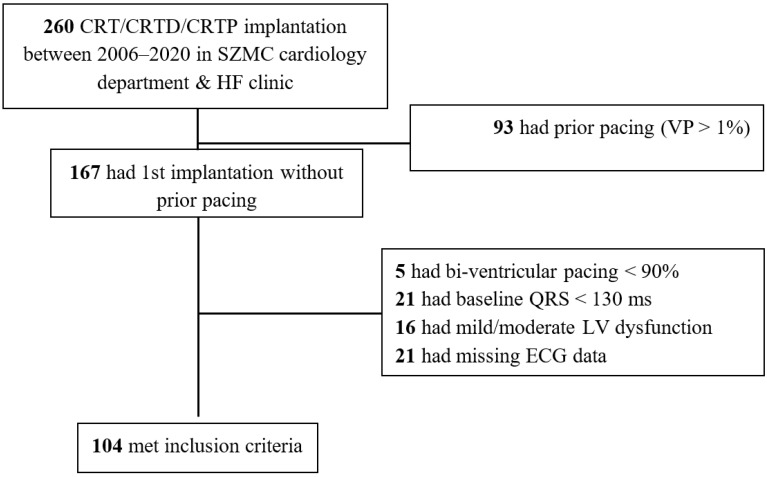
Patient inclusion flowchart.

**Figure 2 jcm-11-01279-f002:**
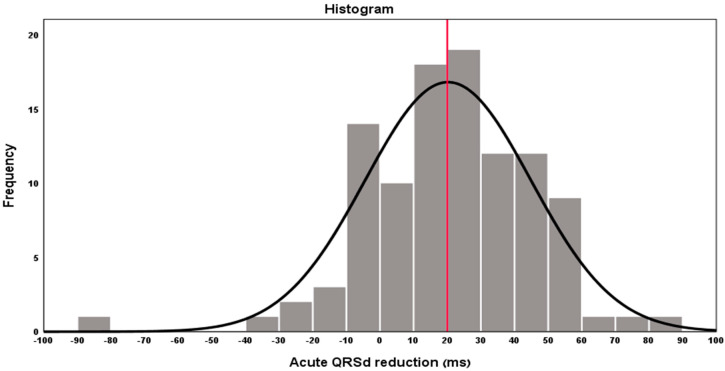
Distribution of Acute QRSd narrowing (ms) in the study population (Red line indicates the mean and median values). Based on the histogram a 20 ms cutoff was set for the definition of acute QRS narrowing. *X*-axis values represent the amount of QRSd reduction, where 0 represents no change in QRSd, negative results represent QRSd prolongation and positive values represent QRSd narrowing after CRT.

**Figure 3 jcm-11-01279-f003:**
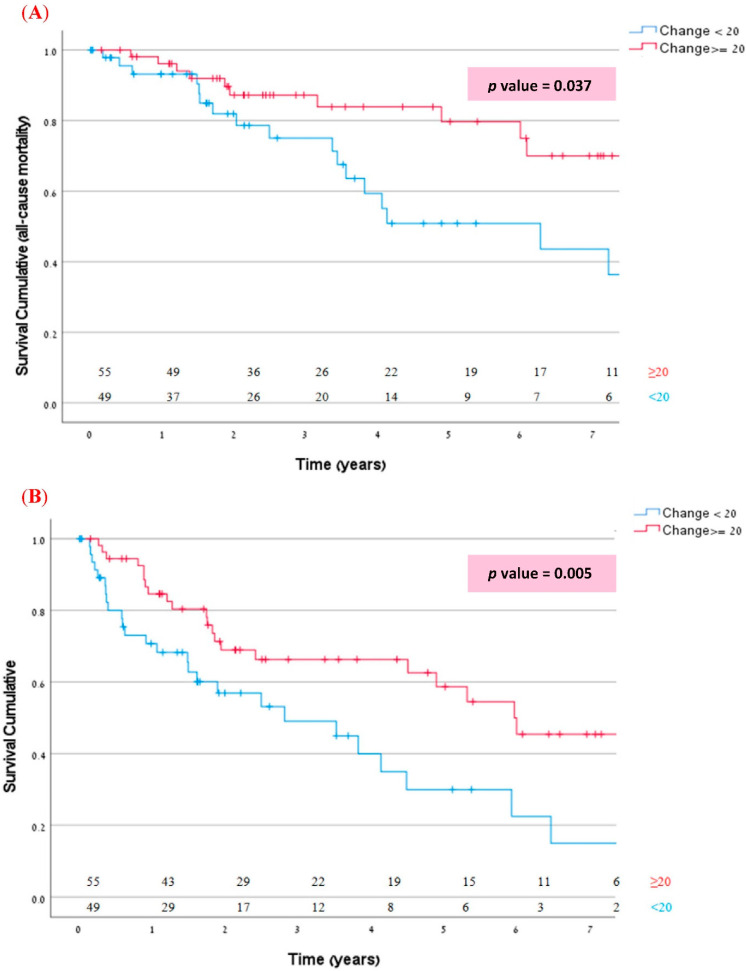
Kaplan-Meir curves for (**A**) mortality and (**B**) Composite of all-cause mortality or HF hospitalization among patients with/without acute QRS narrowing ≥ 20 ms.

**Table 1 jcm-11-01279-t001:** Patients’ characteristics and clinical history prior to CRT implantation according to post-CRT acute QRS narrowing ≥20 ms or <20 ms.

	All (*n* = 104)	Acute QRS Narrowing ≥ 20 ms (*n* = 55)	Acute QRS Narrowing <20 ms (*n* = 49)	Univariate ^1^	Multivariate ^2^	
				*p*-Value	Adjusted OR (95% CI)	*p*-Value
Type of device				1.000 ^3^		
CRTD	98 (94.2%)	52 (94.5%)	46 (93.8%)			
CRTP	2 (1.9%)	1 (1.8%)	1 (2.0%)			
CRT	4 (3.9%)	2 (3.6%)	2 (4.1%)			
Age (Mean ± SD, years)	66.8 ± 11.7	65.8 ± 11.1	68.0 ± 12.3	0.346	0.989 (0.948; 1.031)	0.594
Gender (Female)	26 (25%)	20 (36.3%)	6 (12.2%)	**0.005**	**4.454 (1.521; 13.046)**	**0.006**
CHF etiology (Non-ischemic)	45 (43.3%)	28 (50.9%)	17 (34.7%)	0.096		
HTN	71 (68.3%)	35 (63.6%)	36 (73.5%)	0.282		
DM	46 (44.2%)	22 (40%)	24 (45.0%)	0.357		
HYPERLIPIDEMIA	62 (59.6%)	31 (56.3%)	31 (63.3%)	0.474		
SMOKING	29 (27.9%)	14 (25.4%)	15 (30.6%)	0.558		
Ischemic history						
Prior PCI	57 (54.8%)	25 (45.5%)	32 (65.3%)	**0.042**	0.93 (0.339; 2.87)	0.993
Prior MI	48 (46.1%)	22 (40.0%)	26 (53.1%)	0.182		
Prior CABG	23 (22.1%)	7 (12.8%)	16 (32.7%)	**0.015**	0.414 (0.125; 1.368)	0.148
Prior Valve surgery	14 (13.5%)	6 (10.9%)	8 (16.3%)	0.419		
Prior ICD	4 (3.9%)	1 (1.8%)	3 (6.1%)	0.341 ^3^		
Prior STROKE	8 (7.7%)	2 (3.6%)	6 (12.2%)	0.276 ^3^		
RENAL FAILURE	27 (26%)	12 (21.8%)	15 (30.6%)	0.307		
Admission medications						
B-Blockers	96 (92.3%)	50 (90.9%)	46 (93.9%)	0.719 ^3^		
Diuretics	81 (77.9%)	39 (70.9%)	42 (85.7%)	0.069		
ARB	24 (23.1%)	14 (25.4%)	10 (20.4%)	0.542		
ACEI	59 (56.7%)	30 (54.5%)	29 (59.2%)	0.634		
ARNI	10 (9.6%)	5 (9.1%)	5 (10.2%)	1.000 ^3^		
MRA	75 (72.1%)	40 (72.7%)	35 (71.4%)	0.883		
CCB	5 (4.8%)	2 (3.6%)	3 (6.1%)	0.665 ^3^		
DIGOXIN	15 (14.4%)	5 (9.1%)	10 (20.4%)	0.101		
Anti-coagulation	33 (31.7%)	14 (25.4%)	19 (38.8%)	0.145		
Echocardiographic parameters						
Pre-LV function						
Moderate-Severe				0.488		
Severe	43 (41.4%)	21 (38.2%)	22 (44.9%)			
Fractional shortening	61 (58.6%)	34 (61.8%)	27 (55.1%)			
(Mean ± SD, %)	14.9 ± 4.8	15.0 ± 5.3	14.8 ± 4.2	0.876		
Electrocardiographic parameters						
Sinus Rhythm	91/101 (90.1%)	48/52 (92.3%)	43/49 (87.8%)			
Heart Rate (Mean ± SD, bpm)	72.8 ± 13	73.4 ± 12.7	72.1 ± 13.5			
PR interval (Mean ± SD, ms)	198.4 ± 45.4	192.1 ± 47.2	205.1 ± 42.9	0.518 ^3^		
QTc ^4^ interval (Mean ± SD, ms)	471.6 ± 50.4	471.2 ± 62.6	472.1 ± 34.6	0.636		
Pre-QRSd average	151.6 ± 14.3	156.8 ± 14.5	145.8 ± 11.8	0.184		
(Mean ± SD, ms)				0.931		
LBBB morphology	97 (93.3%)	51 (92.7%)	46 (93.9%)	**<0.001**	**1.068 (1.031; 1.106)**	**<0.001**

^1^ Univariate analysis was done by T-test for continuous variable and Chi-square test for categorical variable. ^2^ Significant variables in the univariate analysis were included in a logistic regression model for multivariable analysis. ^3^ Fisher’s exact test. ^4^ According to Bazett formula. **CRT/CRTD/CRTP** cardiac resynchronization therapy/defibrillator/pacemaker **CHF** chronic heart failure, **HTN** hypertension, **DM** diabetes mellitus, **PCI** percutaneous coronary intervention, **MI** myocardial infarction, **CABG** coronary artery bypass grafting, **ICD** implantable cardioverter defibrillator **ARB** angiotensin receptor blocker, **ACEI** angiotensin-converting-enzyme inhibitor, **ARNI** angiotensin receptor-neprilysin inhibitor, **MRA** mineralocorticoid-receptor antagonist, **CCB** calcium-channel blocker, **LV** left ventricle, HR heart rate, **QTc** QT corrected, **QRSd** QRS duration.

**Table 2 jcm-11-01279-t002:** Predictors of all-cause mortality and mortality or HF hospitalization.

	All-Cause Mortality			All-Cause Mortality or HF Hospitalization		
	Univariante ^1^	Multivariate ^2,a^		Univariante ^1^	Multivariante ^2,a^	
	*p*-Value	HR (95% CI)	*p*-Value	*p*-Value	HR (95% CI)	*p*-Value
Age	**0.004**	1.032 (0.982; 1.086)	0.215	0.104	0.988 (0.950; 1.028)	0.561
Gender (Male)	0.214	0.678 (0.206; 2.230)	0.522	0.158	0.386 (0.144; 1.037)	0.059
CHF etiology (Ischemic)	**0.019**	0.240 (0.019; 3.007)	0.269	**0.006**	1.239 (0.232; 6.625)	0.802
Hypertension	0.703			0.160		
Diabetes mellitus	0.119			**0.021**	1.059 (0.498; 2.252)	0.882
HYPERLIPIDEMIA	0.779			**0.046**	1.508 (0.702; 3.239)	0.292
SMOKING	0.483			0.752		
Ischemic history						
Prior PCI	**0.030**	3.366 (0.535; 21.165)	0.196	**0.012**	1.019 (0.152; 6.833)	0.985
Prior MI	**0.002**	**2.813 (1.217; 6.504)**	**0.016**	**0.002**	**2.770 (1.391; 5.515)**	**0.004**
Prior CABG	**0.002**	1.044 (0.356; 3.059)	0.938	**0.004**	1.104 (0.466; 2.614)	0.822
Prior Valve surgery	0.000	**3.387 (1.352; 8.483)**	**0.009**	**0.001**	2.413 (0.814; 7.151)	0.112
Prior ICD	0.491			0.841		
Prior STROKE	0.048			0.702		
RENAL FAILURE	0.077			0.257		
Echocardiographic parameters						
Pre-LV function (Severe)	0.786			0.253		
Electrocardiographic parameters						
Sinus Rhythm	**<0.001**	0.251 (0.042; 1.484)	0.127	**0.007**	1.589 (0.131; 19.342)	0.716
Heart Rate (bpm)	**0.005**	0.959 (0.921; 0.999) ^b^	0.045 ^b^	0.093		
PR interval (ms)	0.070			**<0.001**	**1.015 (1.008; 1.021)**	**<0.001**
QTc ^4^ interval (ms)	0.399			0.231		
Pre-QRSd average	0.768			0.876		
Acute QRS narrowing	**0.037**	1.922 (0.796; 4.640)	0.146	**0.005**	**3.243 (1.593; 6.603)**	**0.001**
<20 ms						

^1^ Univariate analysis was done by Kaplan-Meir curve for categorical variable and log-rank test for continuous variable. ^2^ Significant variables in the univariate analysis were included in a multivariable model of cox regression. ^4^ According to Bazett formula. ^a^ Multivariable analysis was done while excluding improvement in LV function (due to missing data). ^b^ variable was not included in the multivariable model, but did show statistical significance in entering approach. **CHF** chronic heart failure, **HTN** hypertension, **DM** diabetes mellitus, **PCI** percutaneous coronary intervention, **MI** myocardial infarction, **CABG** coronary artery bypass grafting, **ICD** implantable cardioverter defibrillator, **LV** left ventricle **QRSd** QRS duration.

## Data Availability

The data presented in this study are available on request from the corresponding author. The data are not publicly available due to privacy issues.

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
