# Peer review of "QRS Narrowing Following CRT Implantation: Predictors, Dynamics, and Association with Improved Long-Term Outcome"

_jcm, 2022, doi:10.3390/jcm11051279_

Round 1

Reviewer 1 Report

I read with interest the manuscript by Lapidot et al. In this study, the authors demonstrate that narrowing of the QRS duration within 1 week after CRT implantation is related to long-term mortality and heart failure-induced hospitalizations. Female gender, non-ischemic cause of the cardiomyopathy and baseline QRS duration were shown to be the strongest contributors to the acute QRS change. The manuscript is well-written and well-designed. The Figures are of good quality. I have only a few comments to make.

Major comments:

1) I do miss a critical note on the usefulness of a marker which only provides prognostic info only after the CRT had been implanted. This should be better highlighted in the Discussion.

2) I am intrigued that only 50% of patients show acute QRS narrowing. Were those 50% only those patients with more than 20ms decrease in QRS width, or patients with any decrease in QRS width? In addition, if only 50% show an acute response, is it then a good predictor of long-term response? Typically, circa 70% of CRT recipients do show volume response after CRT (cfr. the study of reference 28).

3) Ideally, missing ECG data on the early and late follow-up moments should be retrieved. The current results on the early and late follow-up ECG's are based on too few data.

Minor comments:

1) The reported mortality is higher than typically reported in CRT cohorts. This needs some mentioning in the Discussion.

2) The same applies for the number of ischemic patients. Circa 50% patients with myocardial infarct is much higher than typically reported.

3) Please check the manuscript for typo's. Some abbreviations are not spelled out fully. Please use either SD or +- when reporting standard deviations, not a mix of both annotations.

Author Response

Reviewer #1:

I read with interest the manuscript by Lapidot et al. In this study, the authors demonstrate that narrowing of the QRS duration within 1 week after CRT implantation is related to long-term mortality and heart failure-induced hospitalizations. Female gender, non-ischemic cause of the cardiomyopathy and baseline QRS duration were shown to be the strongest contributors to the acute QRS change. The manuscript is well-written and well-designed. The Figures are of good quality. I have only a few comments to make.

Answer: We thank the reviewer for his positive review of this manuscript.

Major comments:

  1. A) I do miss a critical note on the usefulness of a marker which only provides prognostic info only after the CRT had been implanted. This should be better highlighted in the Discussion.

Answer: The reviewer raises an important point. Indeed, we describe in the current manuscript a suggested procedural endpoint that might predict long-term outcome rather than a ‘classic’ predictor which should enable to predict which patients would benefit from a procedure and as such should be defined before and not after the procedure. This was written in the discussion (p15 1st paragraph): “Furthermore, although in our study we evaluated QRS width at discharge and not immediately post CRT implant, we suggest >20ms QRS narrowing may serve as a clinical endpoint to define a 'successful' CRT procedure.” Nevertheless, even as a clinical procedural endpoint, this should have been evaluated during the CRT procedure and not after, as the reviewer appropriately remarks. In accordance with the reviewer comments and to highlight these issues we have re-written the relevant discussion paragraph as follows: “Furthermore, although in our study we evaluated QRS width at discharge and not during the CRT implant procedure, we suggest >20ms QRS narrowing may serve as a ‘gold standard’ clinical endpoint for successful CRT procedures. Notably, we present a procedural ‘clinical endpoint’ rather a classic pre-procedural predictor to predict which patients would eventually benefit from CRT procedure. According to our suggested procedural clinical endpoint, an optimal LV lead location during CRT procedure would be a location which will enable biventricular pacing to achieve QRS shortening > 20ms compared to initial pre-CRT QRS width. Anyhow, since QRS shortening in this study was not measured within the CRT procedure, this should be confirmed by future study examining the impact of QRS shortening at the end of CRT procedure on long-term outcome.“ Moreover, we changed the conclusion sentence of the manuscript abstract as follows: “Post-CRT acute QRS narrowing 20ms is independently associated with favorable long-term outcome and might be considered as a novel measure for procedural success", reflecting the limitation resulting from the fact that the marker in the current study was evaluated only after and not during the procedure.

  1. B) I am intrigued that only 50% of patients show acute QRS narrowing. Were those 50% only those patients with more than 20ms decrease in QRS width, or patients with any decrease in QRS width? In addition, if only 50% show an acute response, is it then a good predictor of long-term response? Typically, circa 70% of CRT recipients do show volume response after CRT (cfr. the study of reference 28).

Answer: There were 50% of patients who had QRSd narrowing > 20ms while other 50% were divided into those with < 20ms narrowing, no narrowing, or even QRSd prolongation. This is shown nicely in the QRS reduction histogram- Figure 2. We understand this was not clear enough in the text. Accordingly, we revised the Results paragraph (p10 2nd paragraph) as follows: "A cutoff of 20 ms for the definition of ‘acute QRS narrowing’ was set based on the distribution of acute QRS narrowing among the study population (Figure 2), revealing 55/104 and 49/104 patients with and without acute QRS narrowing >20ms, respectively. Among the 49/104 without QRSd narrowing of >20ms, 25 patients had QRSd narrowing of <20ms and 24 patients had QRSd prolongation." Moreover, we extended the legend of Figure 2 to clearly explain the QRSd histogram results. As seen in Figure 2, 80/104 (77%) of study patients had some QRSd narrowing, which is in line with other studies. As we chose QRS shortening ≥ 20ms as a cutoff to define ‘acute QRS narrowing” in our study, and given the fact that 20ms was the median QRS shortening among our study patients, by definition we had ~50% of patients who fulfilled this criterion.

  1. C) Ideally, missing ECG data on the early and late follow-up moments should be retrieved. The current results on the early and late follow-up ECG's are based on too few data.

Answer: The reviewer is right regarding missing ECG data on Follow up, a limitation acknowledged from the retrospective nature of our study. Nevertheless, since this 'dynamic' nature of QRSd was not reported before, we thought we should still mention this result and mention it in our discussion. In accordance with the reviewer comment we finish the relevant discussion paragraph (p14 2st paragraph) with the following sentence: " However, given the limited number of patients who had regular ECG done early or late post CRT implant, follow-up ECG results need to be validated by future studies. "

Minor comments:

1) The reported mortality is higher than typically reported in CRT cohorts. This needs some mentioning in the Discussion.

2) The same applies for the number of ischemic patients. Circa 50% patients with myocardial infarct is much higher than typically reported.

Answer for both: We accept the reviewer comment regarding the relatively high percent of ischemic patients among our CRT study population. This might also explain some of the increased mortality in our study as ischemic patients were shown to benefit less from CRT device. Accordingly, we added the following sentences to our discussion (p12 1st paragraph): “Notably, around half of our study CRT recipients had ischemic cardiomyopathy, which is higher than reported in other CRT cohorts (25,29), a fact which might have contributed to the relatively high mortality rate among our CRT cohort. “

Nevertheless, we would like to point out that although our mortality is higher than some CRT cohorts (ex. Ref 9, where 186/894 (21%) CRT patients died during 40 month F/U) it has similar mortality to others. For example, Cleland et al in the pivotal CARE HF study (ref 3) had 20% mortality among the CRT cohort during 29 months F/U period. Extrapolating this mortality to the longer 40 months F/U in our study would give a 28% mortality. Similarly, in the study of Jastrzębski et al (ref 28), out of 560 CRT patients, 226 died (40% mortality) during an average of 46 months F/U period.

3) Please check the manuscript for typo's. Some abbreviations are not spelled out fully. Please use either SD or +- when reporting standard deviations, not a mix of both annotations.

Answer: We revised the manuscript for typo's. Regarding SD, we used "+-" throughout the study.

All manuscript text revisions are marked in red.

Reviewer 2 Report

according to your retrospective study results,  in a prospective analysis the 20 msec QRS narrowing during implant should be the "gold standard " parameter to obtain searching optimal sites of LV and RV catheter's final position.  

Author Response

Reviewer #2

  1. According to your retrospective study results, in a prospective analysis the 20 msec QRS narrowing during implant should be the "gold standard" parameter to obtain searching optimal sites of LV and RV catheter's final position.  

Answer: We thank the reviewer for his positive review of this manuscript. In accordance with the reviewer’s comment, we have emphasized the potential impact of our finding as an acute procedural target, or as the reviewer better defines as a ‘gold standard’ parameter for successful CRT procedure. Accordingly, we have revised the last discussion paragraph (p15 1st paragraph) as follows: “Furthermore, although in our study we evaluated QRS width at discharge and not during the CRT implant procedure, we suggest >20ms QRS narrowing may serve as a ‘gold standard’ clinical endpoint for successful CRT procedures…… Accordingly, an optimal LV lead location during CRT procedure would be a location which will enable biventricular pacing to achieve QRS shortening > 20ms compared to initial pre-CRT QRS width.”

We have also emphasized this point in the manuscript conclusion (p16) stating that “We suggest QRS narrowing > 20ms as a possible acute procedural endpoint, to predict improved long-term outcome.”

All manuscript text revisions are marked in red.

Reviewer 3 Report

The authors present a well designed and well written project investigating clinical parameters to predict outcomes post CRT insertion.

They found a QRS shortening of >20ms to independently predict reduced mortality and HF admission.

I can understand the proposed pathophysiological mechanism, but I am unsure as how the 20ms cut-off was chosen. I can see in table 1 that a widened QRS is associated with a greater absolute reduction, but I am surprised in table 2 that the widened QRS did not correlate with outcome (even on univariate analysis).

The paper would be significantly improved if the 20ms cut-off were justified in some manner. Perhaps some data showing a stepwise improvement in outcome with QRR reduction of 0-5, 5-10, 10-15, 15-20, 20-25, 25-30? It would also be important to consider/exclude a %QRS reduction.

I am also surprised that PR interval was a stronger predictor of outcome than QRSd. The authors should more clearly define PR interval. Are they referring to the PR during intrinsic conduction in all cases? Or is it a reflection of the programmed AV delay and hence related to the %Vp? As a related matter the %Vp should be quoted as an electrocardiographic parameter as it predicts outcome.

Author Response

Reviewer #3

A. The authors present a well-designed and well-written project investigating clinical parameters to predict outcomes post CRT insertion. They found a QRS shortening of >20ms to independently predict reduced mortality and HF admission.

Answer: We thank the reviewer for this positive review of our manuscript.

B. I can understand the proposed pathophysiological mechanism, but I am unsure as how the 20ms cut-off was chosen. I can see in table 1 that a widened QRS is associated with a greater absolute reduction, but I am surprised in table 2 that the widened QRS did not correlate with outcome (even on univariate analysis).

The paper would be significantly improved if the 20ms cut-off were justified in some manner. Perhaps some data showing a stepwise improvement in outcome with QRR reduction of 0-5, 5-10, 10-15, 15-20, 20-25, 25-30? It would also be important to consider/exclude a %QRS reduction.

Answer: The reviewer raises a few important issues:

1. Justifying the 20ms QRS shortening cut-off. The 20ms cutoff was defined based on few data which all pointed to this cutoff, including:

a) The distribution of acute QRS narrowing post CRT as stated in the Results section (p10 2nd paragraph): “Mean absolute acute QRS narrowing was 20.2±24.7ms (Median 20.3 ms) and the relative acute QRS narrowing was 12.9±13.5%. A cutoff of 20 ms for the definition of ‘acute QRS narrowing’ was set based on the distribution of acute QRS narrowing among the study population (Figure 2)…”. Thus, this 20ms cut-off was chosen as it was both the mean and median of the acute QRS shortening histogram.

b) Previous publications, in which QRS shortening of 16 ms (ref 28) and 20ms (ref 31) were found to be associated with improved outcome after CRT procedures.

c) We aimed for a simple and clinically practical parameter that may be easy to discern on surface ECG, and will be above the inter-observer error associated with QRSd evaluation, which was found to be around 10 ms (ref 28).

Given the relatively limited number of study patients, analyzing outcome in small subgroups of different QRS duration reduction is not feasible in the current study. In accordance with the reviewer’s comment, we added and emphasized the results of ref studies 28, and 31 to strengthen the rationale supporting 20ms as cutoff.

2. Although QRS duration is a known risk factor for increased mortality among HF patients (based on multiple studies) we did not find it as a significant predictor for adverse outcomes in our study (Table 2). This may be related to the studied population (all receiving a CRT device), whereby a longer QRS duration might have had an improved prognostic impact via QRS duration reduction. Therefore, baseline QRS duration and the QRS reduction post CRT may have had opposing effects on outcome.

3. The reviewer also suggested reporting the data regarding the impact of relative (%) QRS narrowing. Indeed, we have reported acute relative QRS narrowing to be significantly associated with both overall mortality as well as combined endpoint of mortality and HF hospitalization, as written in the Results (p 11, 2nd paragraph), stating that: “In a univariate cox analysis, acute relative QRS narrowing (% of baseline QRSd reduction) as a continuous variable was significantly associated with all-cause mortality (HR 0.974, CI 0.958-0.991, p=0.002, Pearson Correlation=-0.247), and with combined all-cause mortality or HF hospitalization (HR 0.979, CI 0.966-0.992, p=0.002) for every 1% decrease in QRSd.”

C) I am also surprised that PR interval was a stronger predictor of outcome than QRSd. The authors should more clearly define PR interval. Are they referring to the PR during intrinsic conduction in all cases? Or is it a reflection of the programmed AV delay and hence related to the %Vp? As a related matter the %Vp should be quoted as an electrocardiographic parameter as it predicts outcome.

Answer: We examined baseline PR (before CRT implant). To better clarify this point we have revised the definition as “baseline pre-CRT intrinsic PR interval” both in the Results (p11 bottom) and discussion (p 13 1st paragraph). The current manuscript highlights baseline PR as a predictor for CRT outcome and this point is highlighted in the discussion (p 13). However, this is not a novel finding and is not the main focus of the current study.

Regarding %Vp, as we state in the Methods section, we excluded patients with <90% biventricular pacing from this study (p6 bottom paragraph). Thus, we included only patients with biventricular pacing above 90%, considered as clinically effective CRT pacing. Within this small margin (90-100% biventricular pacing) it is unlikely that %Vp will have a significant effect on outcome.

All manuscript text revisions are marked in red.